# The universally-conserved transcription factor RfaH is recruited to a hairpin structure of the non-template DNA strand

**Philipp K Zuber[1], Irina Artsimovitch[2,3], Monali NandyMazumdar[2,3†], Zhaokun Liu[2,3‡], Yuri Nedialkov[2,3§], Kristian Schweimer[1], Paul Rösch[1], Stefan H Knauer[1]\***

[1]Lehrstuhl Biopolymere und Forschungszentrum für Bio-Makromoleküle, Universität Bayreuth, Bayreuth, Germany; [2]Department of Microbiology, The Ohio State University, Columbus, United States; [3]The Center for RNA Biology, The Ohio State University, Columbus, United States

**\*For correspondence:**
stefan.knauer@uni-bayreuth.de

**Present address:** †Department of Genetics and Genome Sciences, Case Western Reserve University, Cleveland, United States; ‡Department of History, Carnegie Mellon University, Pittsburgh, United States; §Division of Pharmaceutics and Pharmaceutical Chemistry, College of Pharmacy, The Ohio State University, Columbus, United States

**Competing interests:** The authors declare that no competing interests exist.

**Abstract** RfaH, a transcription regulator of the universally conserved NusG/Spt5 family, utilizes a unique mode of recruitment to elongating RNA polymerase to activate virulence genes. RfaH function depends critically on an *ops* sequence, an exemplar of a consensus pause, in the non-template DNA strand of the transcription bubble. We used structural and functional analyses to elucidate the role of *ops* in RfaH recruitment. Our results demonstrate that *ops* induces pausing to facilitate RfaH binding and establishes direct contacts with RfaH. Strikingly, the non-template DNA forms a hairpin in the RfaH:*ops* complex structure, flipping out a conserved T residue that is specifically recognized by RfaH. Molecular modeling and genetic evidence support the notion that *ops* hairpin is required for RfaH recruitment. We argue that both the sequence and the structure of the non-template strand are read out by transcription factors, expanding the repertoire of transcriptional regulators in all domains of life.

DOI: https://doi.org/10.7554/eLife.36349.001

## Introduction

NusG/Spt5 proteins are the only transcription factors that coevolved with RNA polymerase (RNAP) since the last universal common ancestor (*NandyMazumdar and Artsimovitch, 2015*). These proteins have an N-terminal domain (NTD) of mixed α/β topology connected to at least one β-barrel C-terminal domain (CTD) bearing a KOW motif *via* a flexible linker. The NTD binds across the DNA-binding channel, bridging the RNAP pincers composed of the β' clamp and β lobe domains and locking elongating RNAP in a pause-resistant state (*Sevostyanova et al., 2011*), a mechanism likened to that of processivity clamps in DNA polymerases (*Klein et al., 2011*). The CTDs modulate RNA synthesis by making contacts to nucleic acids or to proteins involved in diverse cellular processes; *Escherichia coli* NusG binds either to termination factor Rho to silence aberrant transcription (*Mooney et al., 2009b*; *Peters et al., 2012*) or to ribosomal protein S10 to promote antitermination (*Said et al., 2017*) and transcription-translation coupling (*Burmann et al., 2010*).

In addition to housekeeping NusG, diverse bacterial paralogs, typified by *E. coli* RfaH, activate long operons that encode antibiotics, capsules, toxins, and pili by inhibiting Rho-dependent termination, an activity inverse to that of NusG (*NandyMazumdar and Artsimovitch, 2015*). To prevent interference with NusG, action of its paralogs must be restricted to their specific targets. Targeted recruitment is commonly achieved through recognition of nucleic acid sequences, for example, by alternative σ factors during initiation. Indeed, all RfaH-controlled operons have 12-nt operon polarity suppressor (*ops*) signals in their leader regions. RfaH is recruited at *ops* sites in vitro and in vivo

(*Artsimovitch and Landick, 2002*; *Belogurov et al., 2009*) through direct contacts with the non-template (NT) DNA strand in the transcription bubble (*Artsimovitch and Landick, 2002*), a target shared with σ (*Sevostyanova et al., 2008*). However, *E. coli* NusG is associated with RNAP transcribing most genes and lacks sequence specificity (*Mooney et al., 2009a*) arguing against an alternative recognition sites model.

In a working model, off-target recruitment of RfaH is blocked by autoinhibition (*Figure 1*). RfaH-CTD, unlike the CTDs of all other known NusG/Spt5 proteins, which adopt a β-barrel structure, folds as an α-helical hairpin that masks the RNAP-binding site on the NTD (*Belogurov et al., 2007*). Contacts with the *ops* element in the NT DNA are thought to trigger domain dissociation, transforming RfaH into an open, active state in which the NTD can bind to RNAP (*Belogurov et al., 2007*); consistently, destabilization of the domain interface enables sequence-independent recruitment (*Belogurov et al., 2007*; *Shi et al., 2017*). On release, the α-helical CTD spontaneously refolds into a NusG-like β-barrel (*Burmann et al., 2012*), classifying RfaH as a transformer protein (*Knauer et al., 2012*). Activated RfaH remains bound to the transcription elongation complex (TEC) until termination (*Belogurov et al., 2009*), excluding NusG present in 100-fold excess (*Schmidt et al., 2016*). The β-barrel CTD recruits the 30S subunit of the ribosome to leader sequences that lack Shine-Dalgarno elements *via* interactions with S10 (*Burmann et al., 2012*). These interactions could be either maintained throughout translation elongation or broken upon the 70S formation; evidence exists in support of either scenario (*Kohler et al., 2017*; *Saxena et al., 2018*). Following TEC dissociation, RfaH has been proposed to regain the autoinhibited state (*Tomar et al., 2013*), thus completing the cycle.

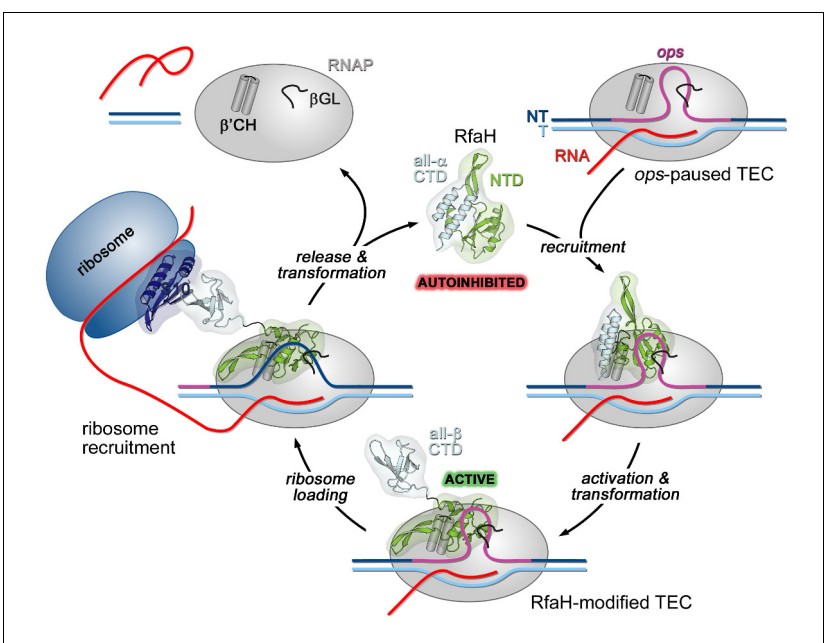

**Figure 1.** Life cycle of RfaH. Available experimental data demonstrate RfaH recruitment to the *ops*-paused RNAP in vitro (*Artsimovitch and Landick, 2002*) and in vivo (*Belogurov et al., 2009*) *via* a hairpin in the NT DNA (this work). *Belogurov et al. (2007)* showed that destabilization of the interdomain interface was required for RfaH switch from the autoinhibited into the active state, and proposed that the RfaH-CTD refolds into a β-barrel upon release. The RfaH-CTD refolding and interactions with S10 were demonstrated by NMR spectroscopy, and functional evidence in support of their role in ribosome recruitment in vivo was reported (*Burmann et al., 2012*). A hypothesis that the autoinhibited state is regained after RfaH is released from TEC at a terminator has been proposed (*Tomar et al., 2013*) and awaits testing. The details of RfaH:RNAP contacts that mediate initial recruitment at *ops*, the molecular mechanism of ribosome recruitment, and hypothetical coupling of transcription and translation by RfaH (*Burmann et al., 2012*) remain to be investigated.β'CH, β' clamp helices; βGL, β gate loop.
DOI: https://doi.org/10.7554/eLife.36349.002

A model of *E. coli* RfaH bound to *Thermus thermophilus* TEC was constructed by arbitrarily threading the NT DNA (absent in the X-ray structure) through the TEC (*Belogurov et al., 2007*). While subsequent functional analysis of RfaH supports this model (*Belogurov et al., 2010*), the path of the NT DNA and the details of *ops*:RfaH interactions remain unclear. The NT DNA is flexible in the TEC (*Kang et al., 2017*) and could be trapped in a state incompatible with productive elongation; RfaH/NusG and yeast Spt5 have been proposed to constrain the NT strand to increase processivity (*Crickard et al., 2016*; *NandyMazumdar et al., 2016*). Direct contacts to the NT DNA have been demonstrated recently for *Bacillus subtilis* NusG (*Yakhnin et al., 2016*) and *Saccharomyces cerevisiae* Spt5 (*Crickard et al., 2016*).

Here we combined structural and functional analyses to dissect RfaH:*ops* interactions. Our data argue that *ops* plays two roles in RfaH recruitment: it halts RNAP to aid loading of RfaH and makes specific contacts with RfaH-NTD. Strikingly, we found that a small hairpin extruded from the NT DNA is required for RfaH recruitment, demonstrating how NT DNA flexibility could be harnessed for transcriptional regulation in this and potentially many other systems.

## Results

### Functional dissection of RfaH:*ops* interactions

Ubiquity of the *ops* sequence in RfaH targets implies a key role in RfaH function. First, *ops* is a representative of class II signals that stabilize RNAP pausing through backtracking, a finding that predates demonstration of direct *ops*:RfaH interactions (*Artsimovitch and Landick, 2000*). Native-elongation-transcript sequencing analysis revealed that *ops* matches the consensus pause signal (*Figure 2A*) and is one of the strongest pauses in *E. coli* (*Larson et al., 2014*; *Vvedenskaya et al., 2014*). The observation that all experimentally validated *E. coli* RfaH targets (*Belogurov et al., 2009*) share a pause-inducing TG dinucleotide (*Chan et al., 1997*; *Vvedenskaya et al., 2014*) at positions 11 and 12 (*Figure 2A*) suggests that delaying RNAP at the *ops* site may be necessary for loading of RfaH. Second, *ops* bases are expected to make specific contacts to RfaH-NTD. However, potential interactions with RfaH are restricted to the central 5–6 nts of *ops* in the NT DNA strand, as these are expected to be exposed on the surface of the *ops*-paused RNAP (*Kang et al., 2017*). Third, binding to *ops* could induce conformational changes in RfaH-NTD that destabilize the interdomain interface to trigger RfaH activation. Finally, pausing at *ops* could be required for ribosome recruitment, a key step in RfaH mechanism (*Figure 1*). In the case of RfaH, pausing could favor 30S loading at sites lacking canonical ribosome binding sites either kinetically or by remodeling the nascent RNA.

To evaluate the roles of individual *ops* bases in vivo we used a luciferase (*lux*) reporter system (*Burmann et al., 2012*) in which RfaH increases expression ~40 fold with the wild-type (WT) *ops* (*Figure 2B*). We constructed reporters with single-base substitutions of all *ops* positions and measured the *lux* activity of the mutant reporters in the presence and absence of ectopically-expressed RfaH. The stimulating effect of RfaH was reduced by every *ops* substitution except for G2C (*Figure 2B*), with the strongest defects observed for substitutions G5A, T6A, G8C, and T11G. Since T11 is buried in the RNAP active site (*Kang et al., 2017*), the strong effect of the T11G substitution is consistent with the essential role of pausing in RfaH activity.

To distinguish between the effects of the *ops* substitutions on RNAP pausing and RfaH binding, we used a defined in vitro system in which RNA chain extension is slowed by limiting NTPs. *Figure 2C* shows assays on the WT, C3G, G5A, and G12C templates, while representative results with all other variants are presented in *Figure 2—figure supplement 1*. The effect of RfaH was determined as ratio of RNA fractions in the presence *vs.* in the absence of RfaH (*Figure 2D*). On the WT *ops* template, RNAP paused at C9 and U11. In the presence of RfaH, pausing at U11 was significantly reduced, but strongly enhanced at G12, a well-documented consequence of RfaH recruitment attributed to persistent RfaH-NTD:DNA contacts (*Belogurov et al., 2007*) and akin to σ-induced delay of RNAP escape from promoters and promoter-like sequences during elongation (*Perdue and Roberts, 2011*). While C3G and T6A substitutions reduced RfaH recruitment and antipausing activity ~3 fold, G4C, G5A, A7T, and G8C abolished both effects completely (*Figure 2D*). Neither of these central bases was required for RNAP pausing (*Figure 2D* and *Figure 2—figure supplement 1*), consistent with their variability in the consensus pause sequence (*Figure 2A*). Conversely, the G12C substitution eliminated the pause at U11, making measurements of RfaH antipausing activity

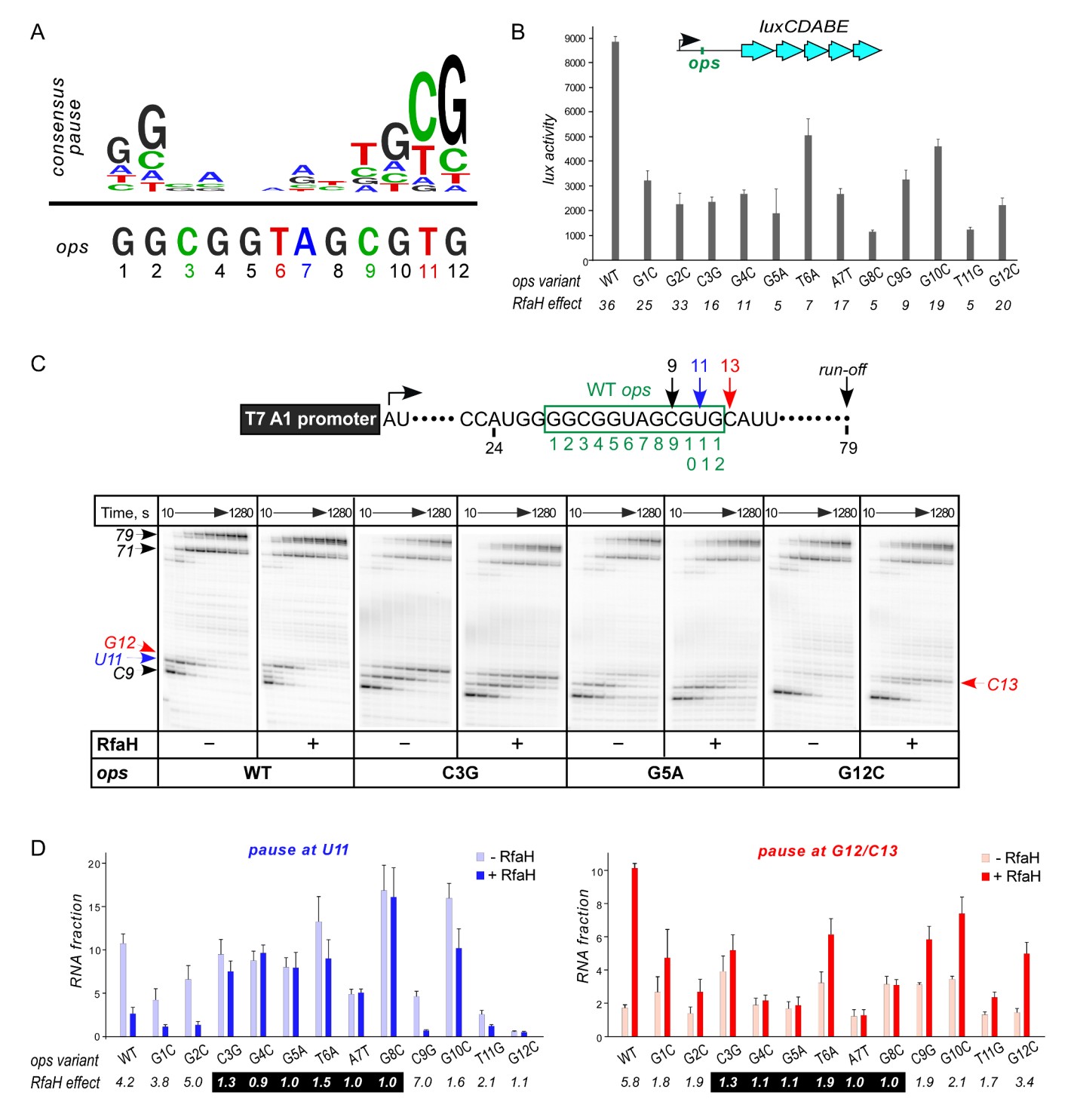

**Figure 2.** Contribution of individual *ops* bases to RNAP pausing and RfaH recruitment. (**A**) Consensus pause and *E. coli ops* sequences. (**B**) Expression of *luxCDABE* reporter fused to *ops* mutants in the absence and presence of RfaH determined in three independent experiments, each with three biological replicates (see source file), is presented as average ± standard deviation. Only the data obtained in the presence of RfaH are plotted; the levels of expression in the absence of RfaH are very low. RfaH effect, the ratio of *lux* activities observed with and without RfaH, is shown below each mutant. (**C**) In vitro analysis of *ops* mutants. Transcript generated from the T7A1 promoter on a linear DNA template is shown on top; the transcription start site (bent arrow), *ops* element (green box), and transcript end are indicated. Halted A24 TECs were formed as described in Materials and Methods on templates with single substitutions in the *ops* element. Elongation was restarted upon addition of NTPs and rifapentin in the absence or presence of 50 nM RfaH. Aliquots were withdrawn at 10, 20, 40, 80, 160, 320, 640, and 1280 s and analyzed on 8% denaturing gels. Positions of the paused and run-

*Figure 2 continued on next page*

*Figure 2 continued*

off transcripts are indicated with arrows. Pause sites within the *ops* region are numbered relative to the *ops* consensus sequence and color-coded. Results with WT, C3G, G5A, and G12C *ops* variants are shown, for all other variants see *Figure 2—figure supplement 1*. (D) Analysis of RfaH effects in vitro (from (C)). The assays were performed in triplicates. RfaH effects at U11 reflect the antipausing modification of RNAP by RfaH. RfaH effects at G12/C13 reflect RfaH binding to the NT DNA strand, which hinders RNAP escape from *ops*. Fractions of U11 RNA (left) and G12 +C13 RNAs (right) at 20 s in the absence or the presence of RfaH, presented as average ± standard deviation from three independent experiments. RfaH effects (determined as a ratio of RNA fractions with *vs.* without RfaH) are shown below the variant. The core *ops* region is indicated by a black box.

DOI: https://doi.org/10.7554/eLife.36349.003

The following source data and figure supplement are available for figure 2:

**Source data 1.** In vivo analysis of *ops* mutants by a *lux* reporter assay.
DOI: https://doi.org/10.7554/eLife.36349.005
**Source data 2.** In vitro analysis of the effect of *ops* mutants on RNAP pausing and RfaH recruitment.
DOI: https://doi.org/10.7554/eLife.36349.006
**Figure supplement 1.** In vitro analysis of *ops* mutants.
DOI: https://doi.org/10.7554/eLife.36349.004

unreliable, but did not abrogate RfaH recruitment (*Figure 2C,D*), suggesting that pausing at U11 is dispensable for RfaH binding when RNAP is transcribing slowly.

Observations that RfaH is recruited to RNAP transcribing the G12C template raised a possibility that recruitment may not be restricted to the U11 position; for example, on this template, RNAP also pauses at the C9 position. To determine whether the entire *ops* element has to be transcribed to recruit RfaH, we assembled TECs on a scaffold in which RNAP is halted three nucleotides upstream from the *ops* site and walked them in one-nt steps to the *ops* pause at U11 (*Figure 3*). To probe RfaH recruitment, we used footprinting with Exo III. In a post-translocated TEC, RNAP protects 14 bp upstream from the RNAP active site (inferred from the position of the RNA 3' end) from Exo III, in a pre-translocated TEC – 15 bp (*Nedialkov and Burton, 2013*). When bound, RfaH alters the trajectory of the upstream DNA duplex to protect additional 6–7 bp of DNA from Exo III (*Nedialkov et al., 2018*). We observed that RfaH induces a strong block to Exo III at U11 (*Figure 3*), as expected based on previous studies (*Artsimovitch and Landick, 2002*). RfaH was also recruited to TECs halted at C9 and G10, but not to G8 TEC in which Exo III was able to digest up to 14 bp of the upstream DNA (*Figure 3*). We conclude that RfaH can bind to TECs halted two nucleotides ahead of the *ops* site. This 'out-of-register' recruitment may be explained by lateral movements of RNAP, which effectively shift the *ops* position (*Figure 3*). In the absence of RfaH, RNAP halted at U11 can backtrack by 2–3 nt and by one nt at G10, whereas C9 TECs are resistant to backtracking (*Nedialkov et al., 2018*); in all three TECs, the same region of the NT DNA will be accessible to RfaH, at least in a fraction of complexes; see Discussion.

## Structural analysis of RfaH:*ops* contacts

Strong effects of substitutions of *ops* bases 3 through 8 on RfaH recruitment but not on RNAP pausing (*Figure 2D*) support a model in which these nucleotides make direct contacts with RfaH. To visualize the molecular details of RfaH:DNA interactions, we determined a crystal structure of RfaH bound to a 9-nt *ops* DNA encompassing bases G2 – G10 (*ops*9) at a resolution of 2.1 Å (*Figure 4A*, *Table 1*). The asymmetric unit contains two molecules of the complex, in which RfaH maintains the closed, autoinhibited state typical for free RfaH (*Figure 4—figure supplement 1A*, (*Belogurov et al., 2007*)). The DNA binds to a basic patch on RfaH-NTD opposite the RNAP/RfaH-CTD binding site and forms a hairpin structure (*Figure 4B*).

The DNA:protein interface encompasses 420 Å$^2$. The hairpin loop comprises G4-A7, with T6 flipped out so that its nucleobase is completely exposed. The other nucleobases of the loop make stacking interactions. Flipped T6 inserts into a deep, narrow, positively charged pocket on RfaH-NTD, which is mainly formed by H20, R23, Q24, and R73 located in helices α1 and α2. G5 packs against the positive surface next to this cavity (*Figure 4B*). RfaH-NTD exclusively contacts nucleotides in the loop region, involving K10, H20, R23, Q24, T68, N70, A71, T72, R73, G74, and V75 (*Figure 4C* and *Figure 4—figure supplement 1B*). Some well-ordered water molecules are located in the *ops*-binding region, but only one participates in the recognition of a base (G4). Base-specific interactions with RfaH-NTD are made by G4, G5, and T6 (*Figure 4C* and *Figure 4—figure*

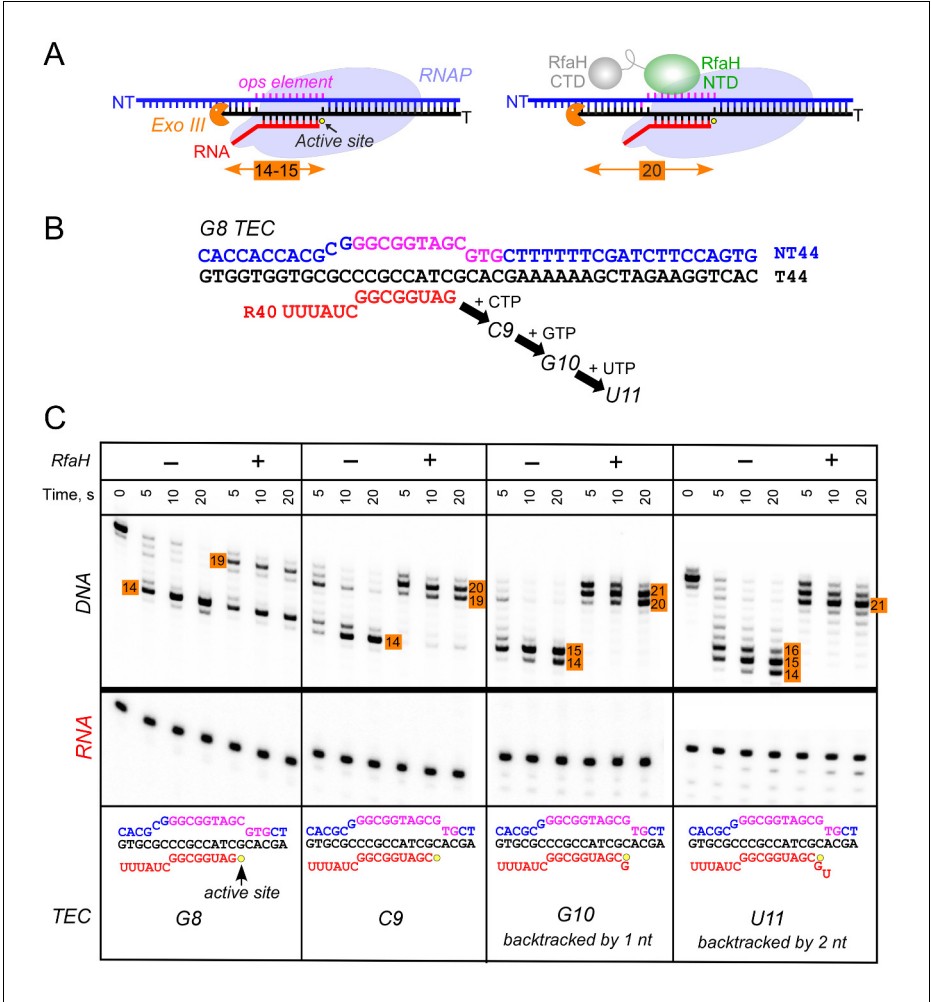

**Figure 3.** RfaH recruitment to RNAP transcribing through the *ops* element. (**A**) Schematic of Exo III footprinting of free and RfaH-bound TECs. Numbers indicate the upstream footprint boundaries relative to the RNA 3' end. (**B**) The G8 TEC was assembled on the scaffold, with RNA and template (T) DNA strands labeled with $[\gamma^{32}P]$-ATP and T4 polynucleotide kinase (PNK), and walked in one-nucleotide steps to C9, G10, and U11 positions in the presence of the matching NTP substrates. (**C**) RfaH was added to 50 nM, where indicated. Following the addition of Exo III, the reactions were quenched at indicated times (0 represents an untreated DNA control) and analyzed on a 12% urea-acrylamide (19:1) gel in 0.5X TBE. Numbers indicate the distance from the RNA 3' end. Hypothetical TEC structures are shown below. G8 and C9 complexes are predominantly post-translocated, as indicated by 14 bp protection of the upstream DNA. In G10 TEC, the pre-translocated state (15 bp protection) is observed, and in U11 an additional backtracked state (16 bp protection). Exo III may counteract backtracking; the sensitivity of the nascent RNA in G10 and U11 TECs to GreB-assisted cleavage (*Nedialkov et al., 2018*) was used to infer the translocation states shown in the schematics.

DOI: https://doi.org/10.7554/eLife.36349.007

supplement 1B); however, only G5 and T6 form a hydrogen-bond network with RfaH-NTD that may underlie sequence-specific recognition. The side chains of K10, H20, R23, and R73 directly interact with the *ops* DNA (*Figure 4C* and *Figure 4—figure supplement 1B*) and no aromatic residues for stacking interactions are located near T6 or G5. Thus, contacts between only two nucleobases and four amino acids mediate specific recognition of *ops* by RfaH. Observations that single Ala substitutions of each RfaH side chain that makes base-specific contacts to G5 and T6 (*Figure 4C*) compromise RfaH recruitment to the *ops*-pausedTEC (*Belogurov et al., 2010*) argue that the RfaH:DNA contacts observed in the binary *ops*9:RfaH complex are functionally important.

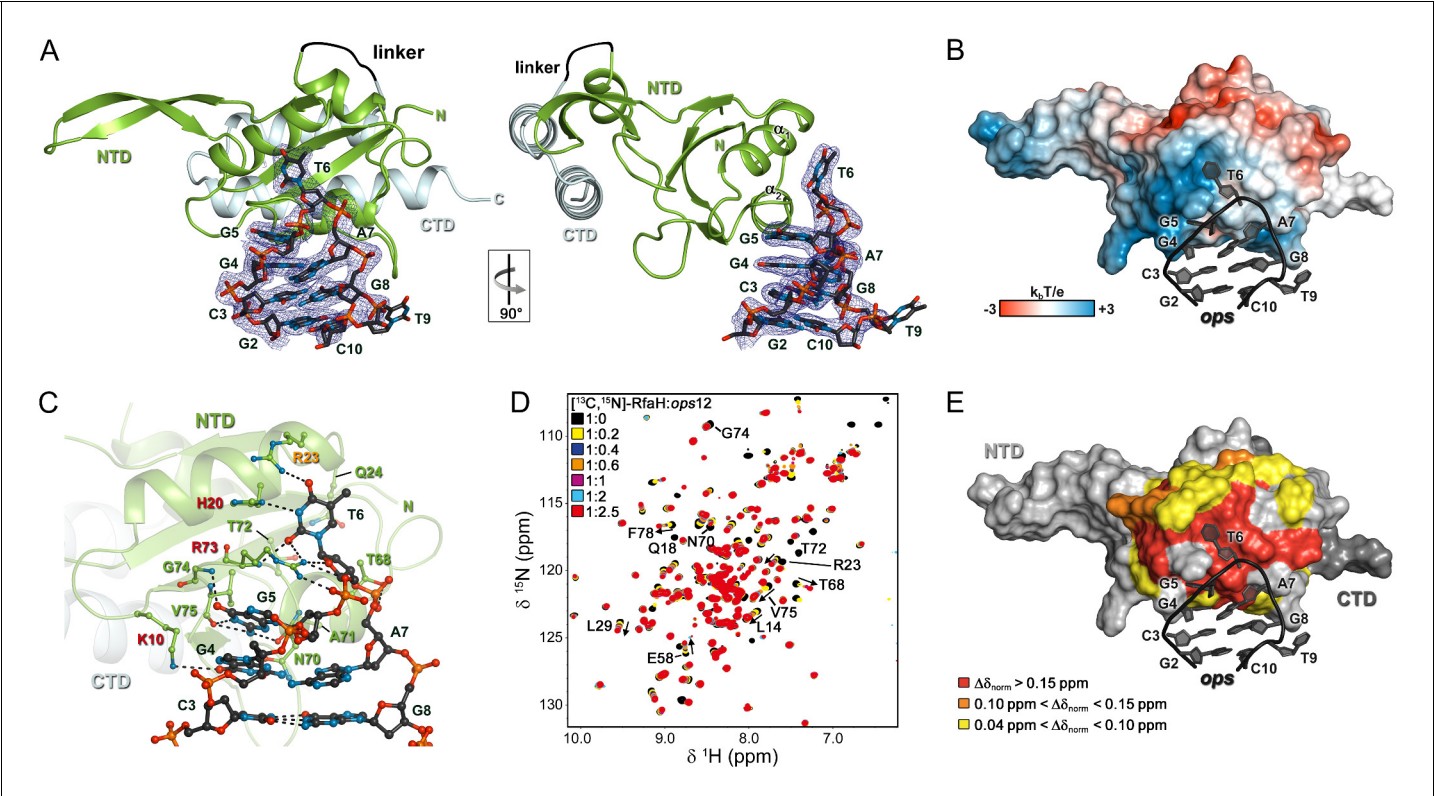

**Figure 4.** Specific recognition of *ops* by RfaH. (**A**) Crystal structure of the RfaH:*ops*9 complex with the $2F_o - F_c$ electron density map contoured at 1 σ. (**B**) Structure of RfaH:*ops*9 complex with RfaH shown in surface representation, colored according to its electrostatic potential and *ops*9 as sticks. (**C**) Details of RfaH:*ops*9 interactions. Hydrogen bonds are shown as black dashed lines. RfaH residues that interact with *ops* are labeled in green. Alanine substitutions of RfaH residues that make base-specific contacts to G5 and T6 *via* their side chains and that compromise RfaH recruitment (*Belogurov et al., 2010*) are highlighted in red (strongly defective) and orange (moderately defective). (**D**) RfaH:*ops* interactions in solution. [$^1$H, $^{15}$N]-HSQC spectra of 110 μM [$^{13}$C, $^{15}$N]-RfaH titrated with 803 μM *ops*12 DNA. Arrows indicate changes of chemical shifts. Selected signals are labeled. (**E**) Mapping of normalized chemical shift perturbations observed in (**D**) on the RfaH:*ops*9 structure.

DOI: https://doi.org/10.7554/eLife.36349.008

The following source data and figure supplements are available for figure 4:

**Source data 1.** Analysis of the chemical shift perturbations during the HSQC-titration of $^{15}$N-RfaH with *ops*12.
DOI: https://doi.org/10.7554/eLife.36349.011
**Figure supplement 1.** Analysis of RfaH:*ops* interactions.
DOI: https://doi.org/10.7554/eLife.36349.009
**Figure supplement 2.** Secondary structure of isolated *ops*9 and RfaH:*ops*9 interaction in solution.
DOI: https://doi.org/10.7554/eLife.36349.010

The stem of the DNA hairpin is formed by base pairs C3:G8 and G2:C10, with T9 being flipped out. The G2:C10 base pair is likely an artifact of crystal packing as the stems of neighboring DNA molecules stack on each other (*Figure 4—figure supplement 1C*) and could not form in a TEC that contains a 10–11 nt bubble. In contrast, the C3:G8 base pair is compatible with the TEC structure and may be physiologically relevant. C3G and G8C substitutions reduce and abolish RfaH recruitment (*Figure 2C,D*), yet these bases lack specific contacts with RfaH (*Figure 4C*), suggesting that a hairpin structure may be necessary.

## The NT DNA hairpin is required for RfaH recruitment

To corroborate the crystallographic data, we carried out solution-state NMR analyses. In the [$^1$H]-NMR spectrum of *ops*9 the single peak at ~13 ppm is characteristic of an imino proton signal of a G or T nucleobase in a DNA duplex, indicating the existence of a hairpin with a single base pair in solution (*Figure 4—figure supplement 2A*). Next, we titrated $^{15}$N-labeled RfaH with WT *ops* (*ops*12)

**Table 1.** Data collection and refinement statistics

| Data collection | |
| --- | --- |
| Wavelength (Å) | 0.9184 |
| Space group | P1 |
| Unit cell parameters | |
| a, b, c (Å) | 36.309/43.187/61.859 |
| α, β, γ (°) | 80.449/75.485/75.392 |
| Resolution (Å)[a] | 41.55–2.1 (2.2–2.1) |
| Unique/observed reflections[a,b] | 19,931/107,345 (2,633/14,210) |
| $R_{sym}$ (%) [a,c] | 6.3 (42.9) |
| $I/\sigma I$[a] | 13.96 (3.47) |
| Completeness (%)[a] | 97.3 (97.9) |
| Molecules per asymmetric unit | 2 |
| Refinement statistics | |
| $R_{work}$ (%)[d] | 18.62 |
| $R_{free}$ (%)[e] | 23.34 |
| Number of atoms | |
| Protein | 4283 |
| Nucleic acid | 574 |
| Water | 116 |
| B-factors | |
| Protein | 56.062 |
| Nucleic acid | 87.427 |
| water | 48.058 |
| r.m.s. deviations | |
| Bond lengths (Å) | 0.013 |
| Bond angles (°) | 1.149 |

[a]Highest-Resolution shell values are given in parentheses.

[b]Friedel mates were not treated as independent reflections.

[c]$R_{sym} = \Sigma_h \Sigma_I \mid I_i(h) - <I(h)> \mid / \Sigma_h \Sigma_i I(h)$; where $I$ are the independent observations of reflection $h$.

[d]$R_{work} = \Sigma_h \mid\mid F_{obs} \mid - \mid F_{calc} \mid\mid / \Sigma_h \mid F_{obs} \mid$.

[e]The free $R$-factor was calculated from 5 % of the data, which were removed at random before the structure was refined.

DOI: https://doi.org/10.7554/eLife.36349.012

and recorded [$^1$H,$^{15}$N]-HSQC spectra after each titration step (**Figure 4D**). Mapping of the normalized chemical shift perturbations (**Figure 4—figure supplement 2B**) on the structure of the RfaH:*ops*9 complex revealed a continuous interaction surface comprising mainly helices α1 and α2 that perfectly matched the DNA-binding site observed in the crystal structure (**Figure 4E**). The signals of $^{15}$N-RfaH-CTD were not affected during the titration, indicating that binding to the *ops* DNA is not sufficient to induce domain dissociation.

The above results demonstrate that base pair C3:G8 forms both in solution and in the crystal of the binary *ops*9:RfaH complex. To evaluate if this hairpin could form in the context of the TEC, we modeled RfaH-NTD bound to the *ops*-paused TEC (**Figure 5A**) based on a recent cryo-EM structure of the *E. coli* TEC (**Kang et al., 2017**) using our *ops*9:RfaH structure. Since NusG and its homologs share the RNAP-binding mode (**Belogurov et al., 2010**; **Bernecky et al., 2017**; **Ehara et al., 2017**; **Said et al., 2017**), the crystal structure of *Pyrococcus furiosus* Spt5 bound to the RNAP clamp domain (**Klein et al., 2011**; **Martinez-Rucobo et al., 2011**) served as a template for modeling. The NT DNA hairpin observed in the *ops*9:RfaH structure could be readily modeled into the TEC. In the modeled complex, RfaH-NTD binds to the β' clamp helices (β'CH) so that the β-hairpin of RfaH,

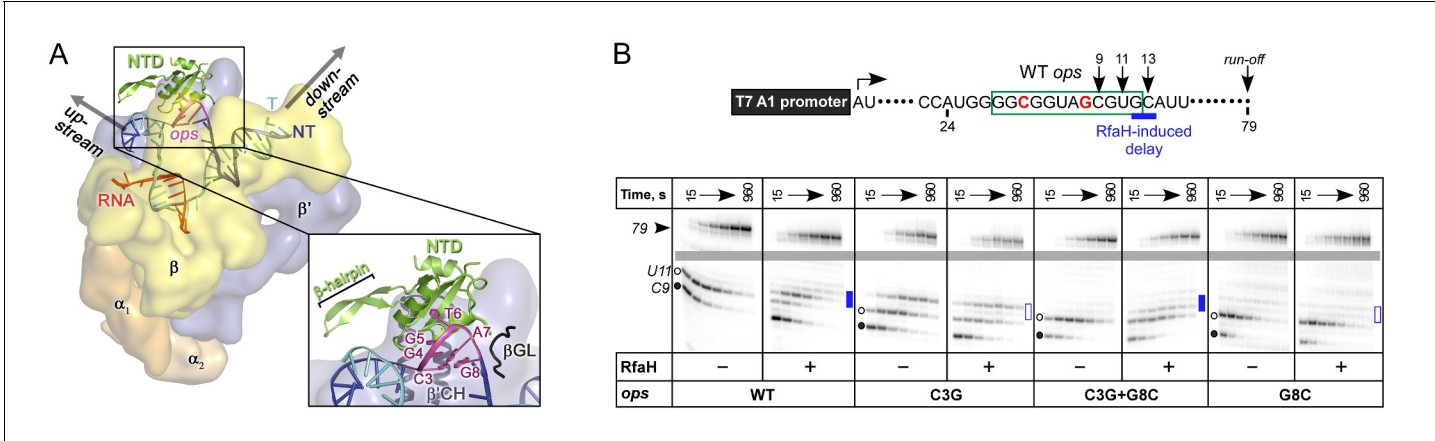

**Figure 5.** The role of NT DNA hairpin. (**A**) Model of RfaH-NTD bound to the *ops*-paused TEC. Surface-accessible NT DNA bases are shown as sticks. (**B**) The double C3G + G8C substitution partially restores RfaH-dependent recruitment. The assay was done as in *Figure 2*. The position of an RfaH-induced delay in RNAP escape is shown with a blue bar, solid if delay is enhanced.
DOI: https://doi.org/10.7554/eLife.36349.013

consisting of β-strands 3 and 4, may establish stabilizing interactions with the upstream DNA, as proposed for *E. coli* NusG-NTD (*Turtola and Belogurov, 2016*).

To test if DNA secondary structure, rather than the identity of the paired nucleotides, is essential for RfaH recruitment to the TEC, we combined strongly defective C3G and G8C substitutions in a flipped G3:C8 base pair. We found that the double substitution partially restored RfaH recruitment, as reflected by RfaH-induced delay at positions 12/13 (*Figure 5B*). We conclude that the C3:G8 base pair (i) can form in the *ops*-paused TEC and (ii) plays an indirect, architectural role in RfaH binding by stabilizing a small DNA loop in which the bases are perfectly positioned to make direct contacts to RfaH-NTD.

## Discussion

### The consensus pause as a versatile regulator

Our findings portray the consensus pause as a chimeric, versatile target for diverse regulatory proteins. Pausing of RNAP is induced by the conserved flanking sequences and would favor recruitment of regulatory factors kinetically, *via* widening the time window for engagement of proteins in low abundance. The central region of the consensus pause is highly variable, and the primary and secondary structures of the surface-accessible NT DNA strand could mediate direct and indirect readout by a protein ligand. We hypothesize that, in addition to RfaH homologs which could be expected to use a similar mode of binding, other unrelated proteins may employ the same general principle during their recruitment to the elongating RNAP. Moreover, contacts with the NT DNA strand that persist after recruitment may underpin regulation of RNA chain elongation in all cells.

### The role of *ops* in RfaH recruitment

Our results confirm that the *ops* element plays several roles in RfaH recruitment. *First*, consistent with the observation of direct contacts with the NT DNA by crosslinking (*Artsimovitch and Landick, 2002*), RfaH interacts with *ops* residues 4 through 7. The interactions are corroborated by previous 'blind', that is, uninformed by the structure, functional studies of RfaH-NTD in which substitutions of RfaH residues that interact with *ops* were found to cause defects in RfaH function (*Belogurov et al., 2010*). However, the pattern of *ops*:RfaH-NTD contacts, and in particular the extrusion of the hairpin, have not been anticipated. We propose that when RNAP pauses at the *ops* site, the NT DNA strand forms a transient hairpin exposed on the surface (*Figures 4* and *5*). Autoinhibited RfaH interacts with the loop nucleotides (G4 through A7), stabilizing the hairpin and forming a transient

encounter complex (*Figure 1*). We observe that T6 flips into a pocket on RfaH-NTD, apparently a common pattern in NT DNA strand contacts since the RNAP σ and β subunits employ analogous capture mechanisms (*Bae et al., 2015*; *Zhang et al., 2012*).

*Second*, pausing at *ops* appears to be required for efficient RfaH recruitment. Substitutions of *ops* residues that reduce pausing compromise RfaH function, even though they do not make direct contacts to RfaH. While the simplest explanation is that pausing simply prolongs the lifespan of the RfaH target, additional roles of pausing could be considered. RNAP backtracks when paused at *ops* in vitro (*Artsimovitch and Landick, 2000*), effectively shifting the exposed NT DNA two nucleotides back. RfaH is recruited to RNAP halted two nts upstream from *ops* (*Figure 3*), suggesting that backtracking at *ops*, assuming it occurs in vivo, may be needed to place the *ops* bases in an optimal position for direct interactions. However, RfaH binds to a scaffold *ops* TEC locked in the post-translocated state (*Nedialkov et al., 2018*), arguing that the NT DNA strand may be sufficiently flexible (*Kang et al., 2017*) to interact with RfaH at several template positions. Although it is also possible that conformational changes that accompany the formation of the paused state may favor RfaH binding to RNAP, recent structures of paused TECs (*Guo et al., 2018*; *Kang et al., 2018*) and our observations that RfaH binds to scaffolds in which the RNA strand is present or missing similarly (*Artsimovitch and Landick, 2002*) do not support this interpretation.

*Third*, given that recruitment of the isolated RfaH-NTD does not require *ops*, we considered a possibility that RfaH contacts to *ops* trigger NTD dissociation from CTD. However, this idea is refuted by our observations that domain interface remains intact in the binary complex, implying that additional interactions with RNAP or nucleic acids relieve autoinhibition. Structural studies of an encounter complex formed when the closed RfaH recognizes *ops* would be required to address this question.

Finally, pausing at *ops* may assist in the recruitment of a ribosome, which is thought to be critical for RfaH-mediated activation of its target genes which lack canonical Shine-Dalgarno elements (*Burmann et al., 2012*). RfaH and NusG make similar contacts to S10 (*Burmann et al., 2010*; *Burmann et al., 2012*) and could bridge RNAP and 30S during translation initiation and 70S during elongation; the *ops*-induced delay could favor the initial RfaH:30S interactions. While a cryo-EM structure of a coupled RNAP:70S complex argues against bridging by NusG or RfaH (*Kohler et al., 2017*), a recent study supports the role of the experimentally determined NusG:S10 interface (*Burmann et al., 2010*) in binding to 70S and transcription-translation coupling in vivo (*Saxena et al., 2018*).

## Specific recognition of *ops* by RfaH

Despite low sequence identity (21% as compared to *E. coli* NusG-NTD), *E. coli* RfaH-NTD has the typical fold of all NusG proteins (*Figure 6A,B*) and is thought to make similar contacts to the β'CH. However, in contrast to sequence-independent NusG, RfaH requires contacts with the *ops* DNA for recruitment. These interactions are highly specific, as illustrated by strong effects of single base substitutions (*Figure 2*) and lack of off-target recruitment in the cell (*Belogurov et al., 2009*). Our present data reveal that the specificity of RfaH:DNA contacts is determined by just a few direct interactions, mediated by a secondary structure in the DNA. We observe that the *ops* DNA forms a hairpin which exposes the invariant G5 and T6, the only two nucleobases that establish a base-specific hydrogen-bond network with RfaH-NTD (*Figure 4C* and *Figure 4—figure supplement 1B*), for specific recognition. In RfaH, the basic patch identified by previous analysis (*Belogurov et al., 2010*) constitutes the DNA binding site, with only the side chains of K10, H20, R23, and R73 making direct contacts to *ops* (*Figures 4B* and *6C*). Alanine substitutions of K10, H20, and R73 dramatically compromised the delay of RNAP escape from the *ops* pause, and thus RfaH recruitment (*Figures 4C* and *6C*, [*Belogurov et al., 2010*]), in agreement with their base-specific interactions in the RfaH:*ops*9 structure. The R16A substitution also had a strong defect (*Belogurov et al., 2010*). However, while one nitrogen atom of the guanidinium group of R16 is in hydrogen bonding distance to the oxygen atom of the G4 base (3.57 Å) in one of the complexes in the asymmetric unit, the distance is larger in the other copy (3.82 Å). Together with the effect of the R16A substitution, this suggests that the R16:G4 interaction may become relevant in the context of the *ops* TEC, where RfaH is more constrained by RfaH:RNAP interactions. Although R23A substitution compromised RfaH recruitment only slightly, our structure reveals that R23 directly contacts T5 *via* its guanidinium group. Q13A, H65A, T66A, and T68A variants showed only mild effects, which may be indirect. Q13 could be

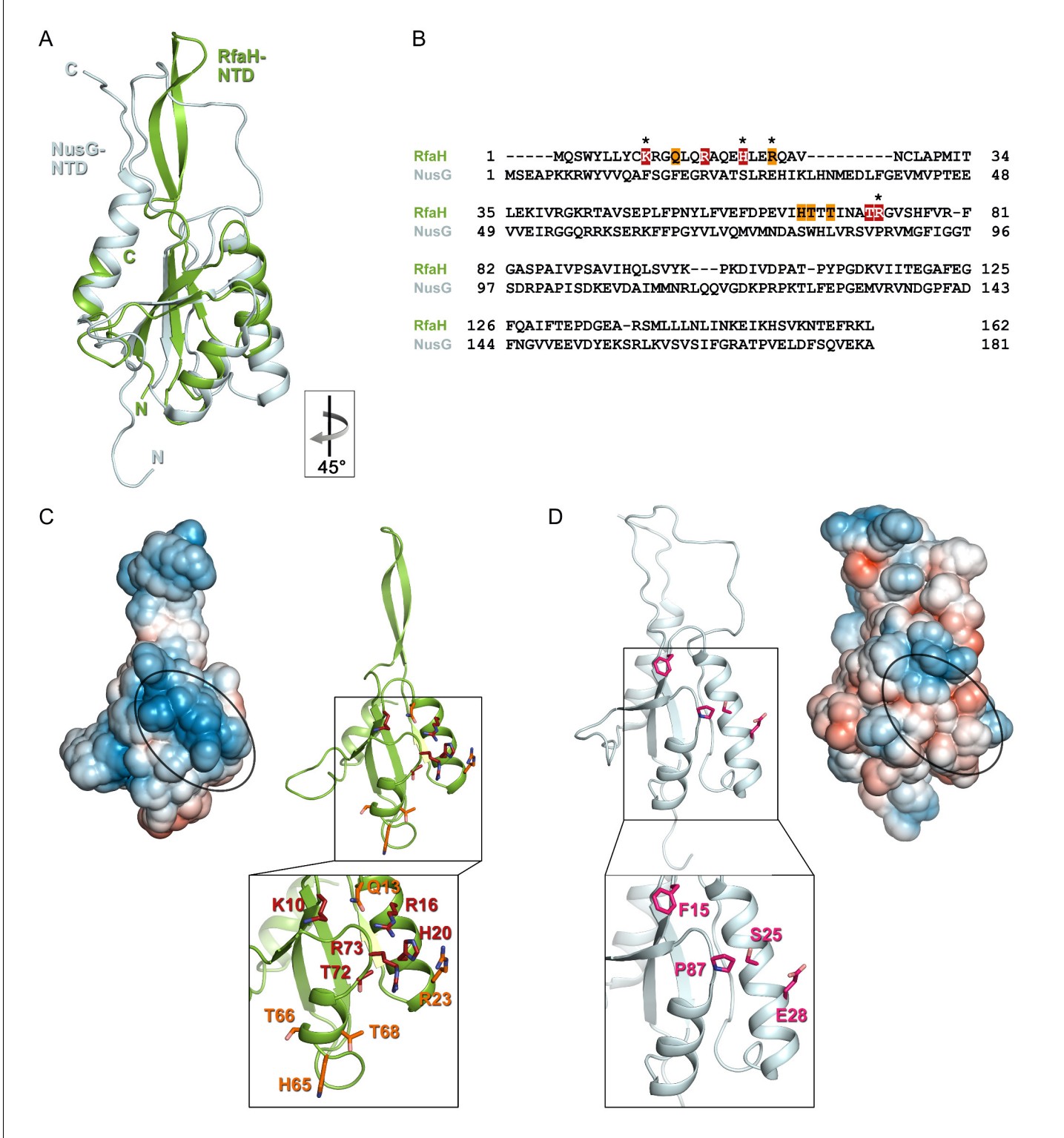

**Figure 6.** Specificity of RfaH for *ops*. Superposition based on backbone atoms of NusG-NTD (PDB ID 2K06, light blue) and RfaH-NTD (taken from the RfaH:*ops*9 structure, green; root mean square deviation: 4.3 Å). Both proteins in ribbon representation. (B) Structure-based sequence alignment of NusG and RfaH. RfaH residues whose substitutions for Ala compromise RfaH recruitment (*Belogurov et al., 2010*) are highlighted in red (strongly defective) and orange (moderately defective). RfaH residues that make base-specific interactions with *ops* via their side chains are marked by an asterisk. (C) Structure of RfaH-NTD in (left) surface representation colored according to its electrostatic potential (from $-3k_BT/e$, red, to $+3k_BT/e$, blue) and (right) ribbon representation with residues highlighted in (B) shown as sticks (C atoms, red or orange; N atoms, blue; O atoms, light red). (D)

*Figure 6 continued on next page*

Figure 6 continued

Structure of NusG-NTD (PDB ID 2K06) in (left) surface representation colored according to its electrostatic potential and (right) ribbon representation. Residues corresponding to the amino acids of RfaH highlighted in (B) are shown as sticks (C atoms, pink; N atoms, blue; O atoms, light red).
DOI: https://doi.org/10.7554/eLife.36349.014

necessary to position R16, while H65, T66, and T68 may be involved in interactions with the β subunit gate loop (*Sevostyanova et al., 2011*). High conservation of K10, H20, R23, and R73 residues (*Shi et al., 2017*) and *ops* sequences (*Belogurov et al., 2009*) suggests a common recognition mechanism for all RfaH proteins.

In contrast, the residues that form the basic patch in RfaH are mostly hydrophobic in *E. coli* NusG (*Figure 6*) and are not conserved within the NusG family (*Shi et al., 2017*), consistent with NusG function as a general transcription factor. However, specific contacts with DNA could explain unusual, pause-enhancing NusG effects on RNA synthesis in some bacteria (*Czyz et al., 2014*; *Sevostyanova and Artsimovitch, 2010*; *Yakhnin et al., 2016*).

## Different read-out modes of the NT DNA strand

The flipping out of T6 in the *ops* element and its insertion into a pocket on RfaH-NTD is reminiscent of a mechanism utilized by σ to recognize the −10 promoter element during initiation (*Bae et al., 2015*; *Zhang et al., 2012*). The melted DNA strand is draped across a positively charged surface of σ, with highly conserved −11A and −7T flipped out into deep pockets of σ, whereas nucleotides at positions −10, −9, and −8 are mainly bound *via* extensive interactions between their sugar-phosphate backbone and σ. In the *ops*9:RfaH complex only one base, T6, is flipped out, but the neighboring G5 packs against the RfaH-NTD surface and also establishes base-specific interactions.

Although both RfaH and σ employ base flipping to specifically bind their target sequences, their recognition mechanisms differ in key details. While the RfaH:*ops* interaction relies only on a very limited number of interactions, σ establishes extensive, base-specific contacts. RfaH exhibits only few interactions with the phosphate backbone and recognizes just two bases specifically, whereas σ makes extensive interactions with the phosphate backbone of the NT DNA strand and establishes base-specific contacts not only with −11A and −7T of the −10 region, but also with −6G of the discriminator element (*Feklistov and Darst, 2011*; *Zhang et al., 2012*). Furthermore, in contrast to RfaH, σ uses a wedge residue (W433 in *E. coli* σ$^{70}$), which rotates into the DNA duplex, mimicking the flipped-out base (*Bae et al., 2015*), a principle that is commonly used by a variety of proteins to stabilize the extrahelical conformation of a flipped-out base (*Davies et al., 2000*; *Lau et al., 1998*; *Yang et al., 2009*; *Yi et al., 2012*). In contrast, RfaH requires that the NT DNA folds into a hairpin to position the two central *ops* nucleotides for specific recognition. The *ops* hairpin thus constitutes an alternative way of stabilizing a DNA conformation with a flipped-out base.

These differences likely reflect distinct roles of NT DNA:protein interactions in the function of RfaH and σ. Although many examples of σ-dependent pauses that are stabilized by σ contacts to promoter-like elements during elongation have been documented (*Perdue and Roberts, 2011*), the primary role of σ is to mediate promoter recognition and DNA melting (*Feklistov et al., 2014*). Interactions with the NT DNA strand are established after initial recruitment to the duplex DNA and are only possible as a result of σ-dependent DNA strand separation. NT DNA:σ interactions are highly specific and utilize the same determinants in promoter and paused complexes (*Marr et al., 2001*; *Zenkin et al., 2007*; *Zhilina et al., 2012*). In contrast, RfaH is recruited to the *ops* element in a pre-made transcription bubble and relies on different DNA contacts for initial binding and for sequence-independent post-recruitment activity. Thus, *ops* recognition by RfaH seems to be more similar to sequence readout by σ during σ-induced promoter-proximal pausing than during promoter melting. Overall, base flipping provides an effective means to read sequence as it allows contacts with all atoms of a base and may be a general mechanism to recruit specific transcription factors throughout transcription.

## The NT DNA strand as a general target for transcription regulation

A growing body of evidence supports a key role of the NT DNA in the regulation of transcription. NT DNA contacts to the β and σ subunits (*Bae et al., 2015*; *Zhang et al., 2012*) determine the

structure and stability of promoter complexes, control start site selection, and mediate the efficiency of promoter escape, in part by modulating DNA scrunching (*Haugen et al., 2006*; *NandyMazumdar et al., 2016*; *Strobel and Roberts, 2015*; *Winkelman and Gourse, 2017*). Upon promoter escape and σ release, the NT DNA loses contacts with RNAP (*Kang et al., 2017*), except for transient interactions with β that control elongation and pausing (*NandyMazumdar et al., 2016*; *Petushkov et al., 2015*; *Vvedenskaya et al., 2014*). Our results suggest that the NT DNA is sufficiently flexible to adopt stable secondary structures and reveal interesting parallels and differences between DNA recognition by σ and RfaH, which bind to similar sites on transcription complexes *via* high-affinity interactions with the β'CH (*Sevostyanova et al., 2008*) and interact specifically with the NT DNA strand *via* base flipping.

NusG homologs from bacteria and yeast that bind NT DNA specifically may employ similar read-out modes, allowing them to exert functions opposing those of *E. coli* NusG (*Crickard et al., 2016*; *Yakhnin et al., 2016*). The available evidence thus suggests that conformational flexibility of the NT DNA and neighboring RNAP elements may produce rich regulatory diversity despite the short length of the exposed NT DNA strand, mediating recruitment of factors that control initiation, elongation, and termination of transcription in all domains of life.

# Materials and methods

**Key resources table**

| Reagent type (species) or resource | Designation | Source or reference | Identifiers |
|---|---|---|---|
| Strain, strain background (*E. coli*) | BL21 (λ DE3) | Novagen | N/A |
| Strain, strain background (*E. coli*) | DH5α Δ*rfaH* (λ DE3) | *Belogurov et al. (2010)* | IA lab stock #149 |
| Recombinant DNA reagent | list of recombinant plasmids used | Table 2 | |
| Sequence-based reagent | *ops*9 GCGGTAGTC | IDT | N/A |
| Sequence-based reagent | *ops*12 GGCGGTAGCGTG | Biomers.net | N/A |
| Sequence-based reagent | T7A1 promoter AAAAAGAGTATTGACTTAAAG TCTAACCTATAGGATACTTAC AGCCATCGAGCAGGCAGCG GCAAAGCCATGG | Sigma Aldrich | IA lab stock #2536 |
| Sequence-based reagent | DN PCR primer AAATAAGCGGCTCTCAGTTT | Sigma Aldrich | IA lab stock #2536 |
| Sequence-based reagent | UP PCR primer AAAAAGAGTATTGACTTAAAG | Sigma Aldrich | IA lab stock #2499 |
| Sequence-based reagent | R40 RNA oligo UUUAUCGGCGGUAG | IDT DNA Technologies | N/A |
| Sequence-based reagent | NT44 DNA oligo CACCACCACGCGGGCGGTA GCGTGCTTTTTTCGATCTT CCAGTG | IDT DNA Technologies | N/A |
| Sequence-based reagent | T44 DNA oligo CACTGGAAGATCGAAAAA AGCACGCTACCGCCCGCG TGGTGGTG | IDT DNA Technologies | N/A |
| Peptide, recombinant protein | *E. coli* RfaH (transcription assays, NMR) | *Belogurov et al. (2007)* | N/A |
| Peptide, recombinant protein | *E. coli* RfaH (crystallization) | *Vassylyeva et al. (2006)* | N/A |
| Peptide, recombinant protein | *E. coli* RNA polymerase | *Svetlov and Artsimovitch, 2015* | N/A |

*Continued on next page*

*Continued*

| Reagent type (species) or resource | Designation | Source or reference | Identifiers |
|---|---|---|---|
| Peptide, recombinant protein | Exo III nuclease | New England Biolabs | Cat#: MO206 |
| Peptide, recombinant protein | T4 polynucleotide kinase | New England Biolabs | Cat#: MO0201 |
| Commercial assay or kit | QIAquick PCR purification kit | Qiagen | Cat#: 28104 |
| Commercial assay or kit | QIAquick Nucleotide Removal Kit | Qiagen | Cat#: 28306 |
| Chemical compound, drug | $(^{15}NH)_4SO_4$ | Campro Scientific | Cat#: CS01-185_148 |
| Chemical compound, drug | D2O | Eurisotop | Cat#: D216L |
| Chemical compound, drug | ApU | Sigma-Aldrich | Cat #: A6800 |
| Chemical compound, drug | $[\alpha-32P]$-CTP | Perkin Elmer | Cat#: BLU008H |
| Chemical compound, drug | Rifapentin | *Artsimovitch et al., 2005* | N/A |
| Chemical compound, drug | PEG monomethyl ether 500 | Sigma-Aldrich | Cat#: 202487 |
| Chemical compound, drug | 4-(2-hydroxyethyl)piperazineethanesulfonic acid (HEPES) for crystallization | Sigma-Aldrich | Cat#: H4034 |
| Chemical compound, drug | MgCl2 for crystallization | Merck | Cat#: 105833 |
| Chemical compound, drug | Glutaraldehyde for crystallization | Fluka | Cat#: 49629 |
| Chemical compound, drug | Tris(hydroxymethyl)aminomethane (Tris) for crystallization | Roth | Cat#: 4855.3 |
| Chemical compound, drug | KCl for crystallization | VWR | Cat#: 26764.298 |
| Chemical compound, drug | Dithiothreitol (DTT) for crystallization | Roth | Cat#: 6908.1 |
| Chemical compound, drug | Perfluoropolyether cryo oil | Hampton Research | Cat#: HR2-814 |
| Software, algorithm | PyMol v. 1.7 | The PyMOL Molecular Graphics System, Schrödinger, LLC. | https://pymol.org/2/ |
| Software, algorithm | COOT | *Emsley et al. (2010)* | https://www2.mrc-lmb.cam.ac.uk/personal/pemsley/coot/ |
| Software, algorithm | XDS | *Kabsch, 2010b* | http://xds.mpimf-heidelberg.mpg.de/ |
| Software, algorithm | XDSAPP | *Sparta et al., 2016* | https://www.helmholtz-berlin.de/forschung/oe/np/gmx/xdsapp/index_e |
| Software, algorithm | PHASER | *McCoy et al. (2007)* | |
| Software, algorithm | PHENIX suite | *Adams et al. (2010)* | https://www.phenix-online.org/ |
| Software, algorithm | LigPlot | *Wallace et al. (1995)* | https://www.ebi.ac.uk/thornton-srv/software/LIGPLOT/ |
| Software, algorithm | NMRViewJ | One Moon Scientific, Inc. | http://www.onemoonscientific.com/nmrviewj |
| Software, algorithm | GraFit v. 6.0.12 | Erithacus Software Ltd. | http://www.erithacus.com/grafit/ |
| Software, algorithm | MatLab v. 7.1.0.183 | The MathWorks, Inc. | https://de.mathworks.com/products/matlab.html |

*Continued on next page*

*Continued*

| Reagent type (species) or resource | Designation | Source or reference | Identifiers |
|---|---|---|---|
| Software, algorithm | ImageQuant | GE Healthcare Life Sciences | www.gelifesciences.com/ |
| Software, algorithm | PISA Server | *Krissinel and Henrick (2007)* | http://www.ebi.ac.uk/pdbe/pisa/ |
| Other | 24-well VDXm plates with sealant | Hampton Research | HR3-306 |

## Plasmids

Plasmids are listed in *Table 2*.

## Gene expression and protein purification

RfaH used in crystallization experiments and in vitro transcription assays was produced as described (*Vassylyeva et al., 2006*), as was RfaH used in NMR experiments (*Burmann et al., 2012*), and RNAP for in vitro transcription assays (*Svetlov and Artsimovitch, 2015*). All expression plasmids are listed in *Table 2*.

The purity was checked by SDS-PAGE, the absence of nucleic acids was checked by recording UV/Vis spectra on a Nanodrop ND-1000 spectrometer (PEQLAB, Erlangen, Germany). Concentrations were determined by measuring the absorbance at 280 nm ($A_{280}$) in a 10 mm quartz cuvette (Hellma, Müllheim, Germany) on a Biospectrometer basic (Eppendorf, Hamburg, Germany).

## Isotopic labeling

$^{15}$N-labeled proteins were obtained from *E. coli* cells grown in M9 minimal medium containing ($^{15}$NH$_4$)$_2$SO$_4$ (Campro Scientific, Berlin, Germany) as sole nitrogen source (*Meyer and Schlegel,*

**Table 2.** Plasmids

| Name | Description | Source |
|---|---|---|
| *ops* variants | | |
| pIA1087 | P$_{BAD}$–*ops*$^{WT}$–*luxCDABE* | *Burmann et al. (2012)* |
| pZL6 | P$_{BAD}$–*ops*(G2C)–*luxCDABE* | This work |
| pZL7 | P$_{BAD}$–*ops*(A7T)–*luxCDABE* | This work |
| pZL12 | P$_{BAD}$–*ops*(T11G –*luxCDABE* | This work |
| pZL14 | P$_{BAD}$–*ops*(G5A)–*luxCDABE* | This work |
| pZL21 | P$_{BAD}$–*ops*(G4C)–*luxCDABE* | This work |
| pZL22 | P$_{BAD}$–*ops*(T6A)–*luxCDABE* | This work |
| pZL23 | P$_{BAD}$–*ops*(G8C)–*luxCDABE* | This work |
| pZL24 | P$_{BAD}$–*ops*(G12C)–*luxCDABE* | This work |
| pZL25 | P$_{BAD}$–*ops*(G1C)–*luxCDABE* | This work |
| pZL26 | P$_{BAD}$–*ops*(C3G)–*luxCDABE* | This work |
| pZL27 | P$_{BAD}$–*ops*(C9G)–*luxCDABE* | This work |
| pZL28 | P$_{BAD}$–*ops*(G10C)–*luxCDABE* | This work |
| pIA1286 | P$_{BAD}$–*ops*(C3G + G8C)–*luxCDABE* | This work |
| Gene expression vectors | | |
| pVS10 | P$_{T7}$ promoter– *E. coli rpoA–rpoB–rpoC*$^{His6}$–*rpoZ* | *Belogurov et al. (2007)* |
| pVS12 | *E. coli rfaH* in pTYB1 | *Vassylyeva et al. (2006)* |
| pIA238 | *E. coli rfaH* in pET28a | *Artsimovitch and Landick (2002)* |

DOI: https://doi.org/10.7554/eLife.36349.015

*1983*; *Sambrook and Russel, 1994*). Expression and purification were as described for the production of unlabeled proteins.

## Crystallization

RfaH was cocrystallized with *ops*9 DNA (5'-GCG GTA GTC-3'; IDT, Coralville IA) based on a published condition (*Vassylyeva et al., 2006*). The protein was dialyzed against crystallization buffer (10 mM tris(hydroxymethyl)aminomethane (Tris)/HCl (pH 7.8), 50 mM KCl, 2 mM DTT). *ops*9 (20 mM in H₂O) was diluted with crystallization buffer and a 5-fold molar excess of $MgCl_2$ before being added to RfaH in a molar ratio of 1:1 (complex concentration 400 µM).

The RfaH:*ops*9 complex was crystallized by vapor diffusion techniques at 4°C using the hanging-drop setup from a reservoir containing 21% (v/v) PEG monomethyl ether (MME) 550, 44.4 mM 4-(2-hydroxyethyl)−1-piperazineethanesulfonic acid (HEPES) (pH 7.0), 4 mM $MgCl_2$ (2 µl protein:DNA solution +2 µl reservoir). Due to crystal instability crosslinking was carried out prior to harvesting by placing 4 µl of 25% (v/v) glutaraldehyde next to the crystallization drop and resealing the well. After an incubation for 2 hr at 4°C the crystal was immersed in perfluoropolyether (Hampton Research) before being frozen in liquid nitrogen.

## Data collection and refinement

Diffraction data were collected at the synchrotron beamline MX-14.1 at Helmholtz-Zentrum Berlin (HZB) at the BESSY II electron storage ring (Berlin-Adlershof, Germany) (*Mueller et al., 2015*) at 100 K using a Pilatus 6M detector and a wavelength of 0.9184 Å. Data were processed and scaled with XDS (*Kabsch, 2010a*; *Kabsch, 2010b*) within the graphical user interface of XDSAPP (*Sparta et al., 2016*). To obtain initial phases Patterson search techniques with homologous search model were performed by PHASER (*McCoy et al., 2007*) using free RfaH (PDB ID 2OUG) as search model. To minimize the model bias a simulated annealing energy minimization using the PHENIX program suite (*Adams et al., 2010*) was performed. Subsequent rounds of model building and refinement were performed using COOT (*Emsley et al., 2010*) and the PHENIX program suite (*Adams et al., 2010*).

## NMR spectroscopy

NMR experiments were performed on Bruker *Avance* 700 MHz spectrometer, which was equipped with a cryo-cooled, inverse triple resonance probe. Processing of NMR data was carried out using in-house routines. 2D spectra were visualized and analyzed by NMRViewJ (One Moon Scientific, Inc., Westfield, NJ, USA), 1D spectra by MatLab (The MathWorks, Inc., Version 7.1.0.183). Measurements involving RfaH were conducted at 15°C, measurements with isolated *ops*9 at temperatures from 4-30°C as indicated. The initial sample volume was 500 µl, if not stated otherwise. The resonance assignments for the backbone amide protons of RfaH was from a previous study (*Burmann et al., 2012*).

The components in the measurement of the ¹⁵N-RfaH:*ops*12 (5'-GGC GGT AGC GTG-3'; biomers.net GmbH, Ulm, Germany) interaction were in 10 mM $K_2HPO_4/KH_2PO_4$ (pH 7.5), 50 mM KCl, 10% $D_2O$. For the determination of the secondary structure of *ops*9 (5'-GCG GTA GTC-3'; metabion international AG, Planegg/Steinkirchen, Germany) the DNA was in 20 mM $Na_2HPO_4/NaH_2PO_4$ (pH 7.0), 50 mM NaCl, 1 mM $MgCl_2$, 10% $D_2O$.

Interaction studies with chemical shifts changes in the fast regime on the chemical shift timescale were analyzed by calculating the normalized chemical shift perturbation ($\Delta\delta_{norm}$) according to *Equation 1* for [¹H,¹⁵N] correlation spectra.

$$\Delta\delta_{\mathrm{norm}} = \sqrt{\left(\Delta\delta^1\mathrm{H}\right)^2 + \left[0.1 \cdot \left(\Delta\delta^{15}\mathrm{N}\right)\right]^2} \qquad (1)$$

where $\Delta\delta$ is the resonance frequency difference in ppm.

## RfaH:*ops* TEC model

The composite model of RfaH bound to the *ops*-paused TEC was generated based on an available cryo EM structure of the *E. coli* TEC (*Kang et al., 2017*) and the complex of *P. furiosus* Spt5 bound to the RNAP clamp domain (*Martinez-Rucobo et al., 2011*). The Spt5:clamp complex was superimposed on the β' subunit of the *E. coli* TEC, and then the RfaH:*ops*9 structure was positioned by

superimposing RfaH-NTD on the NTD of Spt5 using COOT (*Emsley et al., 2010*). Nucleotides 2, 9, and 10 of *ops*9 were manually moved in COOT (*Emsley et al., 2010*) to superimpose with the NT strand keeping the C3:G8 base pair intact so that G2 is the first paired nucleotide on the upstream end of the bubble. The sequence of the remaining *ops* element as well as the corresponding sequences in the T DNA strand and the RNA were adapted.

## Luciferase reporter assays

Luciferase reporter assays were performed as described in (*Belogurov et al., 2010*). A selected *lux* reporter plasmid (*Table 2*) was co-transformed with a plasmid containing the *rfaH* gene (pIA947) or an empty vector (pIA957) into IA149 (Δ*rfaH* in DH5αDE3) and plated on 100 µg/ml carbenicillin (Carb), 50 µg/ml chloramphenicol (Cam) lysogeny broth (LB) plates. Single colonies were inoculated into 3 ml of LB supplemented with Carb and Cam and incubated at 37°C. Overnight cultures were diluted into fresh LB with the antibiotics to optical density at 600 nm ($OD_{600}$) ~0.05 and grown at 37°C for 6 hrs. No induction was required for the $P_{BAD}$-controlled *lux* or $P_{trc}$-controlled *rfaH*, as leaky expression from both these vectors was enough to produce a reproducible signal. Luminescence was measured at approximately equal density for all cultures in triplicates using FLUOstar OPTIMA plate reader (BMG LABTECH, Offenburg, Germany) and normalized for cell density. Three sets of assays were done for each condition, with 3 biological replicates and 6 technical replicates each. We note that low levels of luciferase expression in the absence of RfaH are associated with large errors.

## In vitro transcription assays

Templates for in vitro transcription were made by PCR amplifying pIA1087 (WT *ops*) or the plasmids having *ops* substitutions (*Table 2*) with a T7A1 promoter-encoding primer (5'-AAAAAGAGTA TTGACTTAAAGTCTAACCTATAGGATACTTACAGCCATCGAGCAGGCAGCGGCAAAGCCATGG-3') and a complementary downstream primer (DN: 5'-AAATAAGCGGCTCTCAGTTT-3'). A second PCR was performed with primers 5'-AAAAAGAGTATTGACTTAAAG-3' and DN to reduce the concentration of the unused large primer, followed by purification *via* a QIAquick PCR purification kit (Qiagen, Valencia, CA). The resulting linear templates contained T7A1 promoter followed by an initial 24 nt T-less transcribed region; the run-off transcript generated on these templates is 79-nt long. Linear DNA template (30 nM), holo RNAP (40 nM), ApU (100 µM), and starting NTP subsets (1 µM CTP, 5 µM ATP and UTP, 10 µCi [$\alpha^{32}$P]-CTP, 3000 Ci/mmol) were mixed in 100 µl of TGA2 (20 mM Tris-acetate, 20 mM Na-acetate, 2 mM Mg-acetate, 5% glycerol, 1 mM DTT, 0.1 mM EDTA, pH 7.9). Reactions were incubated for 15 min at 37°C; thus halted TECs were stored on ice. RfaH (or an equal volume of storage buffer) was added to the TEC, followed by a 2 min incubation at 37°C. Transcription was restarted by addition of nucleotides (10 µM GTP, 150 µM ATP, CTP, and UTP) and rifapentin to 25 µg/ml. Samples were removed at time points indicated in the figures and quenched by addition of an equal volume of STOP buffer (10 M urea, 60 mM EDTA, 45 mM Tris-borate; pH 8.3). Samples were heated for 2 min at 95°C and separated by electrophoresis in denaturing 8% acrylamide (19:1) gels (7 M Urea, 0.5X TBE). The gels were dried and RNA products were visualized and quantified using FLA9000 Phosphorimaging System, ImageQuant Software, and Microsoft Excel. In vitro transcription assays were carried out in triplicates and averaged.

## Exonuclease footprinting

To assemble a scaffold TEC, the RNA primer and the T DNA strand were end-labeled with [$\gamma^{32}$P]-ATP using PNK (NEB). Following labeling, oligonucleotides were purified using QIAquick Nucleotide Removal Kit (Qiagen). To assemble a scaffold, RNA and T DNA oligonucleotides were combined in PNK buffer and annealed in a PCR machine as follows: 5 min at 45°C; 2 min each at 42, 39, 36, 33, 30, and 27°C, 10 min at 25°C. 12 pmoles of T/RNA hybrid were mixed with 14 pmoles of His-tagged core RNAP in 30 µl of TB [20 mM Tris-Cl, 5% glycerol, 40 mM KCl, 5 mM MgCl$_2$, 10 mM β-mercaptoethanol, pH 7.9], and incubated at 37°C for 10 min. 15 µl of His-Select HF Nickel Affinity Gel (Sigma Aldrich) was washed once in TB and incubated with 20 µg Bovine Serum Albumin in a 40 µl volume for 15 min at 37°C, followed by a single wash step in TB. The T/RNA/RNAP complex was mixed with the Affinity Gel for 15 min at 37°C on a thermomixer (Eppendorf) at 900 rpm, and washed twice with TB. 30 pmoles of the NT oligonucleotide were added, followed by incubation for 20 min at 37°C, one 5 min incubation with TB-1000 in a thermomixer, and five washes with TB. The

assembled TECs were eluted from beads with 90 mM imidazole in a 15 µl volume, purified through a Durapore (PVDF) 0.45 µm Centrifugal Filter Unit (Merck Millipore), and resuspended in TB. The TEC was divided in two aliquots; one was incubated with 100 nM RfaH and the other with storage buffer for 3 min at 37°C. For each time point, 5 µl TEC were mixed with 5 µl of Exo III (NEB, 40 U) and incubated at 21°C. At times indicated in the *Figure 3* legend, the reactions were quenched with an equal volume of Stop buffer (8 M Urea, 20 mM EDTA, 1x TBE, 0.5% Brilliant Blue R, 0.5% Xylene Cyanol FF).

## Programs

All molecular structures were visualized using The PyMOL Molecular Graphics System (Version 1.7, Schrödinger, LLC.) Superpositions of protein and nucleic acid structures were prepared with COOT (*Emsley et al., 2010*). Interactions between *ops*9 and RfaH were analyzed using LigPlot (*Wallace et al., 1995*). The size of the RfaH:*ops*9 interface was calculated by the PDBePISA server (*Krissinel and Henrick, 2007*).

## Data availability

Coordinates and structure factor amplitudes of the RfaH:*ops*9 complex are deposited in the Protein Data Bank under ID code 5OND.

## Acknowledgements

We thank Angela Fleig and Ramona Heißmann for technical assistance, Birgitta M Wöhrl, Claus Kuhn, and Andrey Feklistov for helpful discussions, and Dmitri Svetlov for comments on the manuscript. We also thank Michael Weyand, Julian Pfahler, and Clemens Steegborn for collecting diffraction data, the HZB for the allocation of synchrotron radiation beamtime and the technical staff of the MX beamline for support. The work was supported by grants Ro 617/21–1 and Ro 617/17–1 (both to PR.) from the Deutsche Forschungsgemeinschaft, and GM67153 (to IA.) from the National Institutes of Health.

## Additional information

### Funding

| Funder | Grant reference number | Author |
| --- | --- | --- |
| Deutsche Forschungsge-meinschaft | Ro 617/21-1 | Paul Rösch |
| National Institutes of Health | GM67153 | Irina Artsimovitch |
| Deutsche Forschungsge-meinschaft | Ro 617/17-1 | Paul Rösch |

The funders had no role in study design, data collection and interpretation, or the decision to submit the work for publication.

### Author contributions

Philipp K Zuber, Writing—original draft, Writing—review and editing, Carried out NMR experiments and analyzed data, Crystallized the RfaH:ops9 complex, Wrote the manuscript; Irina Artsimovitch, Conceptualization, Supervision, Funding acquisition, Writing—original draft, Project administration, Writing—review and editing, Carried out in vitro transcription assays, Designed and supervised in vitro and in vivo experiments, Wrote the manuscript; Monali NandyMazumdar, Carried out in vitro transcription assays; Zhaokun Liu, Carried out luciferase reporter assays; Yuri Nedialkov, Carried out exonuclease foot printing experiments; Kristian Schweimer, Supervision, Writing—original draft, Designed and supervised the NMR experiments, Wrote the manuscript; Paul Rösch, Conceptualization, Supervision, Funding acquisition, Writing—original draft, Project administration, Supervised NMR experiments and crystallographic experiments, Wrote the manuscript; Stefan H Knauer, Conceptualization, Supervision, Visualization, Writing—original draft, Writing—review and editing,

Carried out NMR experiments, Solved the crystal structure of the RfaH:ops9 complex, Generated the RfaH:opsTEC model, Designed and supervised NMR and crystallography experiments, Wrote the manuscript

**Author ORCIDs**
Stefan H Knauer (iD) http://orcid.org/0000-0002-4143-0694

**Decision letter and Author response**
Decision letter https://doi.org/10.7554/eLife.36349.021
Author response https://doi.org/10.7554/eLife.36349.022

## Additional files

### Supplementary files
• Supplementary file 1. RfaH-opsTEC_model.pdb: PDB file of the RfaH:*ops*TEC model.
DOI: https://doi.org/10.7554/eLife.36349.016

• Transparent reporting form
DOI: https://doi.org/10.7554/eLife.36349.017

### Data availability
Diffraction data have been deposited in PDB under the accession code 5OND. All data generated or analyzed during this study are included in the manuscript and supporting files. Source data files have been provided for Figures 2 and 4. The PDB file of the RfaH:ops TEC model has been provided.

The following dataset was generated:

| Author(s) | Year | Dataset title | Dataset URL | Database, license, and accessibility information |
|---|---|---|---|---|
| Zuber PK, Artsimovitch I, Roesch P, Knauer SH | 2018 | RfaH from Escherichia coli in complex with ops DNA | http://www.rcsb.org/pdb/search/structid-Search.do?structureId=5OND | Publicly available at the RCSB Protein Data Bank (accession no. 5OND) |

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
