## [Decision Letter]

Thank you for submitting your article "The universally-conserved transcription factor RfaH is recruited to a hairpin structure of the non-template DNA strand" for consideration by *eLife*. Your article has been reviewed by three peer reviewers, and the evaluation has been overseen by a Reviewing Editor and Gisela Storz as the Senior Editor. One of the three reviewers, Richard Ebridght, has agreed to reveal his identity.

The reviewers have discussed the reviews with one another and the Reviewing Editor has drafted this decision to help you prepare a revised submission.

Summary:

Zuber et al., combined structural and biochemical studies to shed light on the mechanism of the transcription antitermination factor RfaH, a representative of the NusG/Spt5 family of universally conserved factors, which expands the impact of this research. The authors present evidence that RfaH is recruited to a transcription elongation complex (TEC) at an operon polarity suppression (ops) site through a combination of sequence-dependent TEC pausing at ops, and sequence-dependent DNA-hairpin formation and base flipping in the transcription-bubble non-template strand of the TEC paused at ops. They demonstrate that the nt strand of the RfaH-binding site, ops, adopts a hairpin conformation with a flipped-out base, allowing for structure- and sequence-specific read-out of transcription signals. This discovery has implications not just for RfaH mechanism of action, but for our understanding of transcription regulation by signals embedded in DNA as well as mechanisms of recruitment of non-specific regulatory factors, such as DNA helicase UvrD. Conclusions are well supported by the reported NMR, and X-ray crystallographic data, as well as in vivo, and in vitro assays and are also consistent with previous mapping and computational discovery of ntDNA-binding site in RfaH NTD (Belogurov et al., 2010; Svetlov and Nudler, 2012). In conclusion, the manuscript provides important structural and functional insights into the mechanism of transcription regulation by RfaH and related factors, suggesting that sequence-dependent pausing and DNA-hairpin formation can function together to mediate sequence-specific recognition of DNA signals during transcription elongation. The findings should be of broad interest to researchers in transcription and transcriptional regulation. It is thus appropriate for publication once the following suggestions are attended.

Essential revisions:

- The functional importance of the observed interactions for binding affinity and binding specificity (subsection “Structural analysis of RfaH:*ops* contacts”) should be demonstrated experimentally with minimal effort. Although biochemical results are presented to support the inferred DNA secondary structure, no biochemical results are presented to support the inferred protein-DNA interactions. This is an important omission in view of the unusually, potentially worrisomely, small size of the inferred protein-DNA interface: "only 214 A^2.") Just 4 amino acids (K10, H20, R23, and R73) and 2 bases (T6 and G5) are inferred to be important for specificity. Accordingly, a comprehensive analysis -involving substitution of each amino acid by Ala, and quantification of affinities of wild-type and Ala-substituted RfaH derivatives for wild-type, T5-substituted, and G-6-substituted ops derivatives- would be possible.

- The current description of the model for RfaH loading and action is vague (no real Figure legend for Figure 1) and it would be useful to be more concrete both in what is known and what is not known. E.g. in the modelled RfaH-TEC complex, which is very compelling, RfaH is bound to ops in the context of a TEC. The TEC is in a register where U11 would be post-translocated or possibly pre-translocated (assuming G12 has been added). It was believed that the backtracking happens at the U11 pause site (according to Figure 3, Artsimovitch and Landick, 2000) but in that case it is not easy to see how RfaH can bind to ops because the NT DNA should be shifted downstream by at least one, possibly two bases. Do the authors think that RNAP is backtracked at U11? Do the authors think that RfaH binds to this state or do they think it binds to a post-translocated TEC at U11?

- The ExoIII footprints seem to argue against backtracking since the tDNA gets shorter as RNAP progresses towards U11 and the results are not consistent with the schematics at the bottom of Figure 3. It seems that the major ExoIII cleavage products in absence of RfaH are not labelled correctly: G8 should be 14, C9 should be 13, G10 should be 13/12, U11 should be 12. Likewise, in the presence of RfaH, the ExoIII footprints also get shorter (not to the same extent) – major ExoIII cleavage products seem to be 19 for C9, 19/18 for G10 and 18 for U11. These results argue against backtracking because ExoIII gains access to the tDNA strand as RNAP adds bases to the RNA 3'-end. Authors should clarify and discuss this.

---

## [Author Response]

Essential revisions:

- The functional importance of the observed interactions for binding affinity and binding specificity (subsection “Structural analysis of RfaH:ops contacts”) should be demonstrated experimentally with minimal effort. Although biochemical results are presented to support the inferred DNA secondary structure, no biochemical results are presented to support the inferred protein-DNA interactions. This is an important omission in view of the unusually, potentially worrisomely, small size of the inferred protein-DNA interface: "only 214 A^2.") Just 4 amino acids (K10, H20, R23, and R73) and 2 bases (T6 and G5) are inferred to be important for specificity. Accordingly, a comprehensive analysis -involving substitution of each amino acid by Ala, and quantification of affinities of wild-type and Ala-substituted RfaH derivatives for wild-type, T5-substituted, and G-6-substituted ops derivatives- would be possible.

We currently do not have an assay that would allow us to measure RfaH affinities to the EC directly and quantitatively; an EMSA with ^32^P-labeled RfaH that we used in (Artsimovitch and Landick, 2002) is messy and semi-quantitative. We use an indirect way to assess defects in recruitment by measuring RNAP retention 1 and 2 nucleotides downstream from the *ops* site (referred to as 12 and 13 in the current manuscript). We assume that the strength of RfaH contacts to the NT strand is reflected in the delay in RNAP escape from the *ops* site, but this is not a true affinity measurement. Moreover, only a fraction of RNAP is delayed downstream from the *ops* site. Using this assay, we showed that K10A, H20A, and R73A substitutions had strong defects in delaying RNAP escape from *ops*, whereas R23A had a moderate effect (Belogurov et al., 2010). Based on these results, we proposed that K10, H20, R23, and R73 directly interact with the *ops* DNA.

In light of the reviewers’ concerns about the small size of the RfaH:*ops*9 interaction surface, we recalculated the interface using the PISA server. We found that the interface comprises 420Å^2^ (average of both complexes in the asymmetric unit), a value updated in the revised manuscript. The crystal structure and the NMR data presented in the current work and a cryoEM structure of RfaH:*ops* EC obtained in the Darst lab support this interpretation and reveal additional contacts that were not predicted by modeling.

We have already demonstrated the importance of RfaH K10, H20, R23, and R73 (Belogurov et al., 2010) and G5 and T6 *ops* residues (this work) in RfaH function in vitro and in vivo and we do not yet have an assay to provide adequate quantitative analysis of RfaH binding to its real target, the EC. Thus, we cannot properly carry out a thorough analysis of these interactions in a reasonable time frame.

We agree that a systematic analysis of RfaH:EC interactions would be necessary to validate structural observations, particularly because the details of RfaH:DNA contacts are not identical. However, this cannot be achieved with “a minimal effort” because roles of all determinants, not just RfaH K10, H20, R23, R73 and G5/T6 in *ops*, need to be probed, and because we need to develop an equilibrium binding assay to probe contributions of RfaH and DNA residues implicated by X-ray and cryoEM structures.

At the same time, a manuscript reporting the cryoEM structure of RfaH-*ops* EC is expected to be accepted any day now. This manuscript was submitted five months after the results of the current study have been communicated to the authors and went through review at Cell with a lightning speed. Given these developments, we cannot afford a significant delay in publication. Notably, the two manuscripts are complementary because the Darst manuscript, on which IA and YN are coauthors, does not present any functional data on the NT DNA determinants and conformation.

To better describe what is known about RfaH:DNA interactions, we revised the text and figures to summarize the available functional data about the RfaH:DNA contacts. We indicated the demonstrated effects of RfaH substitutions on recruitment to *ops* (Belogurov et al., 2010) in Figure 4C and Figure 6 and discussed the observed effects of RfaH substitutions in the light of our RfaH:*ops*9 complex structure.

- The current description of the model for RfaH loading and action is vague (no real Figure legend for Figure 1) and it would be useful to be more concrete both in what is known and what is not known. E.g. in the modelled RfaH-TEC complex, which is very compelling, RfaH is bound to ops in the context of a TEC. The TEC is in a register where U11 would be post-translocated or possibly pre-translocated (assuming G12 has been added). It was believed that the backtracking happens at the U11 pause site (according to Figure 3, Artsimovitch and Landick, 2000) but in that case it is not easy to see how RfaH can bind to ops because the NT DNA should be shifted downstream by at least one, possibly two bases. Do the authors think that RNAP is backtracked at U11? Do the authors think that RfaH binds to this state or do they think it binds to a post-translocated TEC at U11?

Backtracking by 1 and 2-3 nt is observed (by Gre-assisted cleavage) in ECs stalled by substrate deprivation G10 and U11 ECs, respectively, in the absence of RfaH; no backtracking is seen in C9 EC. RfaH inhibits backtracking if added to the preformed ECs because it stabilizes the -10 bp at the upstream fork junction, which has to melt for backtracking to occur, as inferred from an increase in inter-strand DNA crosslinking by psoralen. These data are from a manuscript that has been just accepted for publication by Nedialkov et al.

We assume that in vivo RfaH is recruited at U11 because a strong pause at this position was observed by permanganate footprinting (Leeds and Welch, 1996) and later by NET seq, and because mutations that interfere with RNAP pausing at U11 compromise RfaH function, but we do not know whether this complex is pre-, post-, hybrid- (Guo et al., 2018 and Kang et al., 2018), or reverse-translocated. Because RfaH binds to scaffolds in which the RNA strand is present or missing similarly (Artsimovitch and Landick, 2002), it is likely that RfaH primarily senses the NT DNA conformation during the initial recruitment and binds to the EC at different positions and in different translocation states, owing to the inherent flexibility of the NT DNA. RfaH binds to the U11 *ops* EC which is backtracked but also to the *ops* EC in which both RNA and T DNA are post-translocated in the cryoEM structure of RfaH:*ops*EC; in the latter complex, RNA translocation is forced by the scaffold topology, and the NT DNA forms a hairpin with the C3:G8 bp.

We modified the text and the legend of Figure 1 to provide more details on what is and what is not known about the life cycle of RfaH. We also expanded the discussion of RfaH recruitment in the revised manuscript.

- The ExoIII footprints seem to argue against backtracking since the tDNA gets shorter as RNAP progresses towards U11 and the results are not consistent with the schematics at the bottom of Figure 3. It seems that the major ExoIII cleavage products in absence of RfaH are not labelled correctly: G8 should be 14, C9 should be 13, G10 should be 13/12, U11 should be 12. Likewise, in the presence of RfaH, the ExoIII footprints also get shorter (not to the same extent) – major ExoIII cleavage products seem to be 19 for C9, 19/18 for G10 and 18 for U11. These results argue against backtracking because ExoIII gains access to the tDNA strand as RNAP adds bases to the RNA 3'-end. Authors should clarify and discuss this.

We apologize for a cryptic description of the ExoIII data. We used ExoIII as a reporter of RfaH recruitment: an extension of ExoIII upstream protection is due to RfaH effects on the NT DNA and the upstream DNA duplex trajectories. RfaH induces a strong block at C9, G10, and U11 ECs, consistent with its recruitment at these positions. In Figure 3, ExoIII footprint boundaries are labeled with respect to the RNA 3’ end; in a completely post-translocated EC, RNAP protects 14 upstream bp from ExoIII (9 bp RNA:DNA hybrid + 5 extra bp), in a half- translocated (Guo et al., 2018 and Kang et al., 2018) or pre-translocated EC– 15 bp. G8 EC, in which the upstream RNA is not complementary to the T DNA, is post-translocated (the reviewer is correct and this number should have been 14). C9 is also post-translocated, the T DNA becomes 1 nt shorter, and only 1 predominant species is observed. In G10, even both 14- and 15- products are seen, whereas in U11, an additional backtracked (16) species is seen. Exo III may not be an ideal probe because once ExoIII cleaves the DNA, RNAP cannot go back. We infer the translocation register in these complexes from sensitivity to Gre cleavage (Nedialkov et al., 2018), but the Exo probing also shows that RNAP is reluctant to move forward as it approaches *ops*.

We expanded the section on ExoIII probing and modified Figure 3 and its legend to make it more clear.